# A nanophotonic laser on a graph

Michele Gaio[1], Dhruv Saxena [1], Jacopo Bertolotti [2], Dario Pisignano[3,4,5], Andrea Camposeo[3] & Riccardo Sapienza [1]

Conventional nanophotonic schemes minimise multiple scattering to realise a miniaturised version of beam-splitters, interferometers and optical cavities for light propagation and lasing. Here instead, we introduce a nanophotonic network built from multiple paths and interference, to control and enhance light-matter interaction via light localisation. The network is built from a mesh of subwavelength waveguides, and can sustain localised modes and mirror-less light trapping stemming from interference over hundreds of nodes. With optical gain, these modes can easily lase, reaching ~100 pm linewidths. We introduce a graph solution to the Maxwell's equation which describes light on the network, and predicts lasing action. In this framework, the network optical modes can be designed via the network connectivity and topology, and lasing can be tailored and enhanced by the network shape. Nanophotonic networks pave the way for new laser device architectures, which can be used for sensitive biosensing and on-chip optical information processing.

[1] The Blackett Laboratory, Department of Physics, Imperial College London, London SW7 2AZ, UK. [2] Physics and Astronomy Department, University of Exeter, Stocker Road, Exeter EX4 4QL, UK. [3] NEST, Istituto Nanoscienze-CNR, Piazza San Silvestro 12, 56127 Pisa, Italy. [4] Dipartimento di Matematica e Fisica "Ennio De Giorgi", Universitá del Salento, via Arnesano, 73100 Lecce, Italy. [5] Dipartimento di Fisica "Enrico Fermi", Universitá di Pisa, Largo B. Pontecorvo 3, 56127 Pisa, Italy. These authors contributed equally: Michele Gaio, Dhruv Saxena  Correspondence and requests for materials should be addressed to R.S. (email: r.sapienza@imperial.ac.uk)

Network science describes complex systems with a focus on the interaction between the elementary units, looking beyond the microscopic details[1]. This has proven fruitful in many different fields, to understand the way a virus spreads[2], quantum information is transmitted[3], or a power grid failure propagates[4]. Likewise in nanophotonics, a network approach is found to be beneficial for understanding and enhancing emission, scattering, and concentration of light in complex photonic systems[5–7]. Nevertheless, conventional nanophotonic architectures, for example integrated circuits[8] or nanophotonic lasers[9], mostly avoid multiple scattering. While this clearly simplifies the photonic design, it misses out on harnessing the potential of multiple scattering and interference to engineer optical modes. For example, in nanophotonic networks formed by interconnected optical waveguides, the complex and multiple interference of the many recurrent loops can lead to topology-dependent mode formation, as shown in a different context by quantum graph theory[10], with high quality factors or a specific spectral profile. A low-dimensional multiple scattering architecture can also enhance the interaction of light and promote coupling of emitters, due to spatial confinement in quasi-one-dimensional or two-dimensional systems[11–13]. In addition, in a photonic network, light scattering occurring at the nodes and long-distance light transport are decoupled, which enables the scattering strength to be designed via the connectivity (i.e., the number of links) and light transport to be designed via the link length and network size, which provides an added design advantage.

A photonic network with an embedded gain medium can enhance the probability of stimulated emission and so provide a novel lasing cavity. Moreover, the optical modes formed by recurrent scattering and interference, can be designed by the network topology. In a disordered material, multiple and recurrent scattering boosts the stimulated emission probability, resulting in efficient random lasing[14,15] and in a wealth of interesting physical phenomena[12,16,17]. In the context of photonic networks, random lasing has been studied in a perforated membrane with subwavelength network-like topology[18] albeit without light confinement in the links. Very recently a macroscopic laser based on a few connected optical fibres was demonstrated (four nodes), as a first network laser operated in the single scattering regime[19]. Networks have also been proposed for optical routing[13,20] and for light localisation[21].

Here, we introduce a planar nanophotonic network fabricated from a mesh of nanoscale waveguides, forming a micron-scale

photonic material comprising over 200 nodes. We report experimental observation of network lasing exhibiting low threshold and evidence of light localisation in the photonic network. To rationalise these results, light propagation and interference in these systems is described with a graph model valid in the multiple scattering regime. Modelling complex lasing systems with a graph theory approach[22,23] capable of reducing its complexity is still largely unexplored. In nanophotonics, most scattering media, for example semiconductor powders[24] or perforated membranes[18], are usually described either with an independent scattering or coupled-dipole approximation[25], or with direct numerical solution to the Maxwell's equations[26]. A graph approach to light propagation in complex materials offers instead a simple and highly effective framework to describe realistic-sized systems in an accurate way, and a powerful tool to design random lasing action.

## Results

**Random lasing in a nanophotonic network.** Our nanophotonic networks consist of subwavelength waveguides (links) connected together at their crossing (nodes), as shown in Fig. 1a. They are fabricated by electrospinning single-mode polymer nanofibres (diameters in the range 200–500 nm) doped with a laser dye (see Methods). Each structure is a planar disordered network with a partial mesh topology, where all nodes connect directly and non-hierarchically to their nearest-neighbour. The network has degree $D$, defined as the number of connections of each node but excluding the ones on the periphery, in the range 4 to 6 (average degree is 4.4), average link length $\langle l_e \rangle = 26.5\,\mu m$, and around 200 nodes, as shown in Fig. 1c, d. The network topology is similar to that formed by overlapping randomly oriented needles, as in the Buffon's needle experiment[27], however our network is more connected as the nanofibres are joint together at the nodes (see Methods). Each subwavelength nanofibre guides light with a propagation length larger than 100 μm, as measured previously[13], and the gain length $\ell_g$ is estimated to be 10 μm, which is calculated from the stimulated emission cross-section of the dye by $\ell_g = (\text{density} \times \text{cross-section})^{-1}$ (see Supplementary Note 7).

The photonic network is very efficient to reach room-temperature lasing when excited with a single 500 ps green laser pulse ($\lambda = 532\,nm$), which illuminates a spot of diameter ~160 μm in the network plane. Optical images of fluorescence and lasing are shown in Fig. 1a, b. Under low excitation intensity ($P = 20\,nJ$) the outcoupled fluorescence from the fibres is

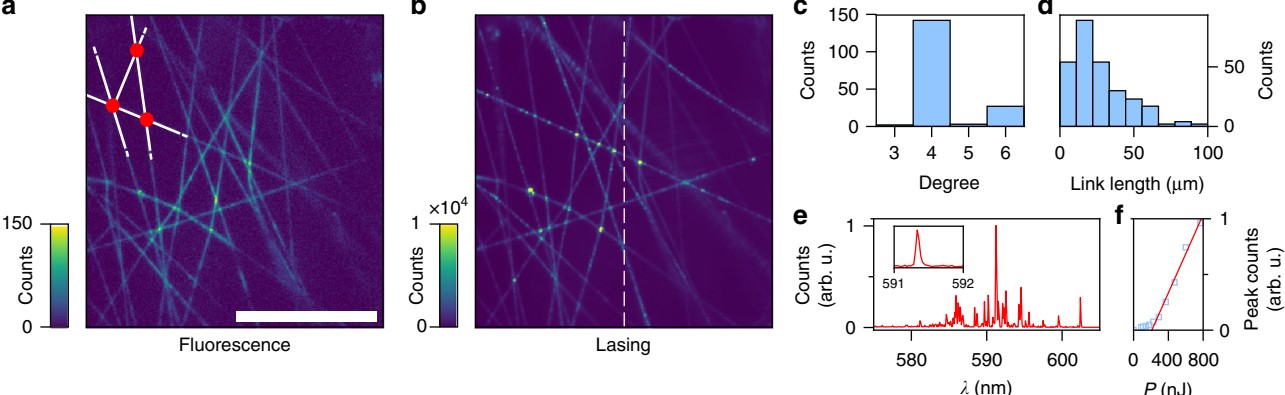

**Fig. 1** Random lasing in a photonic network. Far-field images of **a** fluorescence and **b** lasing from a network of free-standing subwavelength electrospun polymer nanofibres embedded with dye (Rhodamine 6G). Scale bar is 100 μm. The white lines and red circles in the top left of image (**a**) illustrate the links and nodes of the network, respectively. The network topology is characterised by its node degree distribution (**c**) and link length distribution (**d**). The lasing spectrum (**e**) collected along the vertical dashed line in **b** reveals multimodal behaviour and narrow linewidth (inset: laser peak at threshold). **f** The plot of peak emission intensity as a function of pump power shows a clear transition to lasing at threshold $T \simeq 200\,nJ$

recorded (Fig. 1a) and its pattern follows the network shape. Note that spontaneous emission couples to the fundamental waveguide mode in these subwavelength fibres with an efficiency as high as 30–50%[13]. When the illumination intensity is increased, particularly above the lasing threshold, bright spots corresponding to the network nodes are observed (Fig. 1b, $P = 2000$ nJ). This is because stimulated emission populates the guided modes, which are outcoupled at the nodes due to out-of-plane scattering.

Spectrally, lasing is characterised by highly multimode emission with sharp peaks, as shown in Fig. 1e, with only a minor pulse-to-pulse variation (see Supplementary Fig. 3). The experimental linewidth is limited by the spectrometer resolution of 0.05 nm (see Methods). The threshold is around 200 nJ, i.e. ~1 mJ cm$^{-2}$, as extracted from the emitted peak power versus pump intensity relation shown in Fig. 1f. It is worth noting that the data in Fig. 1 is representative of the sample; in total, data was collected from 16 different networks and the threshold ranged between 120–325 nJ.

Spatially, the lasing modes are embedded in the network and therefore hard to access with far-field measurements. Nevertheless, we can infer their extension from the light that scatters out of the network. Figure 2a–c show the scattered light at three specific lasing peaks (indicated by circles in Fig. 2d), with a bandwidth of 0.1 nm, measured by an imaging spectrometer (with Schmidt corrector). To obtain these hyperspectral images, the sample plane was imaged and scanned over the 50 µm input slit of the spectrometer by translating the imaging lens, as shown in Fig. 2e (see Methods). The dashed circles in Fig. 2a–c indicate the pump regions. The lasing light is coupled into the network and is scattered out from regions several 100 µm far from the pump region. Moreover, different modes occupy different sets of network links, as evident by comparing the spatial extent of different modes in Fig. 2a–c (see Supplementary Fig. 4). This is an indication that the lasing modes are not delocalised over the full sample.

Further proof of the localised nature of the modes can be obtained from the statistics of the nearest-neighbour level spacing. The level spacing is a robust experimentally accessible quantity that relates the modes with the transport regime[28]. Diffusive and localised systems are known to follow universal nearest-neighbour level spacing statistics[29] regardless of the details of the system, as predicted for instance in the random matrix framework[30]. Broad spatial overlap and mode coupling of delocalised modes induces mode repulsion[31] resulting in the Wigner-Dyson distribution in the nearest-neighbour level spacing, whereas localised modes lack spatial overlap and therefore feature a random spectrum with no mode repulsion, and are well-described by a Poissonian nearest-neighbour level spacing distribution[32,33]. The level spacing obtained from our experimental lasing spectra is shown in Fig. 2f, and follows an exponential trend indicative of localisation. The dip at smaller spacing is due to the finite resolution of our spectrometer (0.05 nm). For comparison, we also overlay the Wigner-Dyson distribution (dotted lines). We note that random matrix theory description is valid for a passive system[34] and so the mode interaction above threshold could lead to a change in the lasing spectrum, and therefore of the mode spacing statistics. However, we operate close to the lasing threshold, and as shown in Supplementary Fig. 2, the lasing peaks do not shift in frequency with pump power.

**Photonic modes on a graph.** Network random lasing can be well-described by a graph model, which we develop to study the key elements of the nanophotonic network beyond its microscopic details. We solve Maxwell's equations (scalar wave equation) on a metric graph, in a way similar to that done for quantum graphs[35] (see Methods). The links at the boundary of the network are left open and represent the only loss channels in the modelled system. Within the graph model, the full spatial profile of the modes can be easily calculated without the need

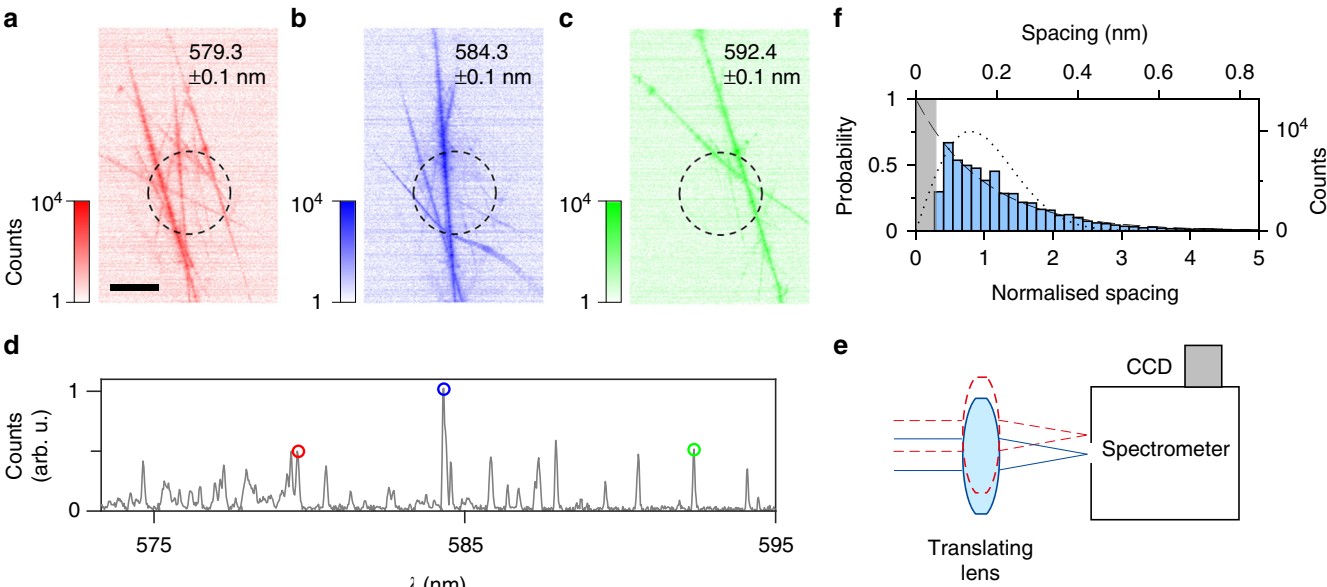

**Fig. 2** Characterisation of network lasing modes. Far-field images of the lasing network corresponding to three different lasing modes, at wavelengths: **a** 579.3 ± 0.1 nm, **b** 584.3 ± 0.1 nm and **c** 592.4 ± 0.1 nm. The dashed black circle indicates the pump area. Scale bar is 100 µm. **d** Spectral data from the entire area shown in **a**. Coloured circles indicate the lasing peaks corresponding to the hyperspectral images (**a–c**). Hyperspectral images were obtained by scanning the lens in front of the spectrometer (**e**) and collecting spectral data from different areas of the sample. **f** Normalised mode spacing statistics obtained from experimentally measured lasing spectra from different parts of the sample. The distribution is well fitted by an exponential curve (dashed line) up to the experimental spectral resolution indicated by the grey band. Wigner-Dyson distribution (dotted line) is also shown for comparison

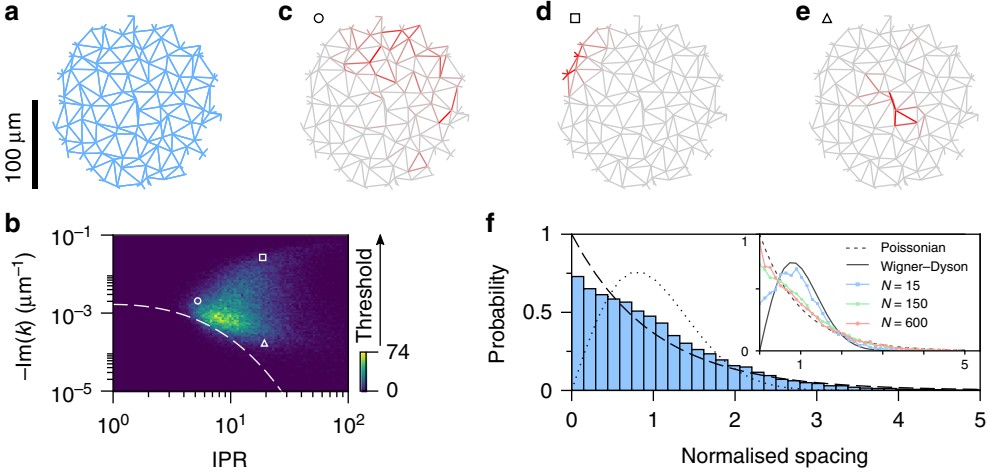

**Fig. 3** Properties of the network modes. **a** For a given network topology (143 nodes, 293 links, average degree $D = 6$ (excluding nodes at the periphery), average edge length $\langle l_e \rangle = 6.5\,\mu m$), the spatial profile and complex $k$ of the optical modes can be computed. **b** Modes are characterised by the lasing threshold ($-\mathrm{Im}(k)$) and a degree of localisation described by inverse participation ratio (IPR). The white dashed line is a prediction for modes localised at the centre of the network. **c–e** The type of mode in different areas of the plot (indicated by the corresponding symbol) is confirmed by inspecting their spatial profile (intensity in red): (**c**, circle) delocalised modes occupying a significant system area; (**d**, square) lossy modes confined close to the network boundary; (**e**, triangle) modes localised in the centre of the system, which are those with the lowest threshold. **f** The level spacing statistics obtained from the model complex wavevectors' ($k$), for networks with 150 nodes and topology same as in **a**. The distribution follows mostly an exponential trend with limited mode repulsion. The mode spacing distribution for smaller and larger networks (15 and 600 nodes) is shown for comparison in the inset

for spatial discretisation. This enables efficient modelling of large networks. Although a similar approach was recently developed[19], it was limited to the single scattering process. Instead, here we focus on multiple scattering in the optical mesoscopic regime, by considering complex networks with a large number of nodes (200–600 nodes, Fig. 3a). Note that numerical methods such as finite-difference time-domain (FDTD) would be impractical for modelling such large systems and providing extended statistics.

The networks modes that are determined from our model are labelled with complex wavevector $k$. Under the assumption of linear and undepleted gain we can calculate the mode threshold by evaluating $-\mathrm{Im}(k)$, which is the amount of gain required to bring the mode to net amplification. This approach is common to many lasing models[36,37], and is valid for pump energies not far from the lasing threshold, which is the common assumption for the first lasing modes.

A direct correlation between the degree of localisation and lasing threshold is observed in the plot of lasing threshold, i.e., $-\mathrm{Im}(k)$ versus inverse participation ratio (IPR), as shown in Fig. 3b. The IPR is defined as $\mathrm{IPR} = L \int dx |E|^4 / \left( \int dx |E|^2 \right)^2$ and is normalised to range between 1 (for uniformly delocalised modes) and $\sim N$ (where $N$ is the total number of links in the network) for modes predominantly confined to a few links. The modes localised at the centre of the network, with a radial spatial profile following $\exp(-r/\xi)$, are expected to lie along the dashed white line shown in Fig. 3b; $\mathrm{IPR} \simeq (R/2\xi)^2$, $-\mathrm{Im}(k) \propto R \exp(-R/\xi)/2\xi^2$, where $R$ is the radius of the network and $\xi$ is the localisation length (see Supplementary Note 3 for derivation). In Fig. 3b two main branches are visible, one follows the above predicted trend, with lower threshold modes being characterised by stronger confinement, while another branch seems to have the opposite behaviour with larger threshold for the most localised modes. Upon inspection of their spatial profiles, the localised modes with high threshold are found to be localised close to the network boundary, as in Fig. 3d, and so are subject to higher losses. This is therefore simply a finite-size effect and such modes are still

capable of lasing albeit with larger thresholds. The localised modes confined in the middle of the network, as in Fig. 3e, i.e., with the lowest in-plane outcoupling losses, are the ones with the lowest threshold, and so are the likely lasing modes.

The graph analysis allows us to evaluate the mode spacing distribution, by considering the modes obtained numerically from the model. The resulting distribution is shown in Fig. 3f and also follows an exponential trend, as observed experimentally (Fig. 2f). Although slight mode repulsion is still visible in Fig. 3f, this arises from the finite-size of the simulated network. Upon simulating larger networks, the level spacing distribution clearly converges towards an exponential behaviour, as shown in the inset of Fig. 3f, when the network is larger than the mode localisation length.

**Threshold control with topology.** We further use our graph model to investigate the role of network topology on the lasing action. Specifically, we consider the effect of the network degree on the lasing threshold, for degree $D = 3, 4, 5, 6$. For a given number and density of scattering points, changing the connectivity of the system has two main effects: firstly it affects the scattering at the nodes, as light is distributed among different number of links, and secondly it modifies the linear length of the network, thereby changing the total gain available. The simulation in Fig. 4a shows how the network degree impacts the lasing threshold. A rapid decrease of the lasing threshold for increased connectivity is visible, with a drop of threshold by an order of magnitude between $D = 3$ and $D = 6$ ($T_{D=3}/T_{D=6} = 10 \pm 0.2$), as shown in Fig. 4b. These threshold values ($T$) are obtained from the threshold distributions by computing all the modes in a given wavelength window ($\Delta\lambda = 1\,\mathrm{nm}$ centred at $\lambda_0 = 600\,\mathrm{nm}$) and calculating the average of the lowest threshold modes over 1000 different realisations. We point out that the increase in network length with increasing degree, which is an increase in the total gain available, cannot explain the threshold decrease alone. The total gain increases by a factor of $\sim 2$ (when comparing $D = 3$ and $D = 6$) while the threshold reduces by a factor of $\sim 10$.

**Sensitivity**. The localised modes of the networks are very sensitive to local perturbations, either in the form of non-uniform pumping or as induced by a local change in refractive index, mimicking the binding of a target analyte. Figure 5 describes these two effects. In Fig. 5a, b, a non-uniform pump profile, with the shape of a checkerboard, modifies the lasing profile by suppressing or enhancing a few lasing modes. This experimental result is due to spatially dependent modal gain, which favours a mode against another and is very efficient because of mode localisation[38,39].

Moreover, the network is very sensitive to a local change of the refractive index. In Fig. 5c, d we plot how the modes of the network shift when the refractive index of a single link is changed. For a change as small as $\Delta n = 10^{-2}$, we can see a clear shift in some of the lasing modes, those that are localised around that specific link. This gives us a refractive index sensitivity of 20 nm per RIU (refractive index units) or 1 nm per RIU per micrometre

(link length 20 μm). In comparison, a global change in the refractive index across all links of the network (total length 4.834 mm) has a sensitivity of 570 nm per RIU (see Supplementary Fig. 5) or 0.12 nm per RIU per micrometre. The network is thus ten times more sensitive to local perturbations than global changes in refractive index.

## Discussion

Nanophotonic networks provide a conceptually new device architecture for designing and fabricating chip-compatible random lasers. In particular, the process of light scattering (at the nodes) and transport (in the one-dimensional links) are decoupled, and so can be independently designed via the network topology. Nanofabrication methods such as near-field electro-spinning[40], direct ink writing[41] and soft lithographies[42] would in principle allow for highly controlled deposition of the individual waveguides, with pre-defined and independently controlled connectivity and size. Furthermore, a network architecture can enhance light-matter interaction and promote loop formation and coupling between embedded emitters, due to its low dimensionality, which is favourable for lasing. Although three-dimensional (3D) bulk arrangements of polymer nanofibres have been demonstrated[43–47], previously reported architectures were weakly connected or consisted of self-connected loops, and therefore lasing was mostly sustained by long fibres. This is very different from our connected network, where the gain length ($\ell_g \sim 10\,\mu m$) is of the order of the link length, therefore single edge lasing is impossible, and efficient lasing requires multiple scattering and more than ~50 μm wide illumination. Compared to 3D arrangements, planar architectures are also better suited for on-chip integration with other photonic components.

The graph approach we have proposed to model light transport and localisation in the network is computationally efficient and correlates well with the experimental results, even though it is limited by the assumption of single-mode links, which is satisfied only for subwavelength link diameter, and neglects out of network scattering due to impurities or node morphology. The out-of-plane scattering at the nodes are of the order of a few percent, depending on the geometry at the node, as confirmed by FDTD

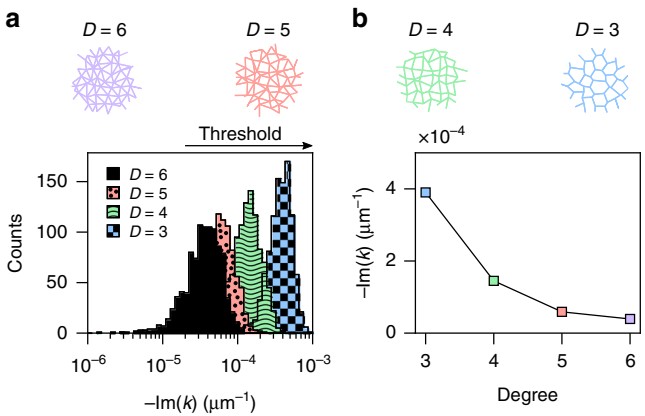

**Fig. 4** Degree and threshold relation from model. **a** Threshold ($-\mathrm{Im}(k)$) distributions for networks with average degree $D = 3,4,5,6$ and constant number and density of nodes ($N = 600$). **b** The lowest lasing threshold estimated from the model (averaged over 1000 different realisations) as a function of the average degree

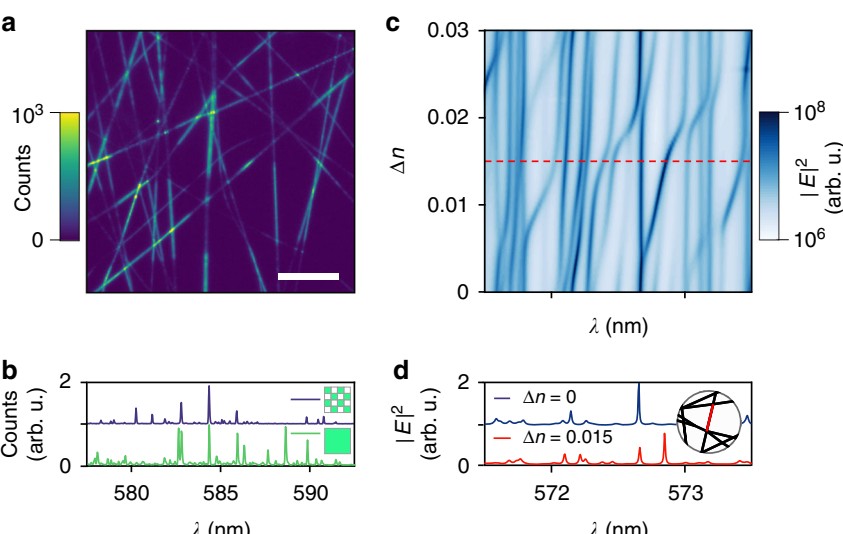

**Fig. 5** Spectral shaping and optical sensing. **a** Far-field image showing fluorescence from a network that is pumped with checkerboard pattern, with patches of size of 52 × 60 μm. Scale bar is 100 μm. **b** Measured lasing spectrum obtained when pumping the network with the checkerboard pattern (top), as well pumping it uniformly (bottom). **c** Calculated map of the lasing mode shift versus the refractive index shift of a selected link. A few modes shift, and mode coupling is evident in the avoided crossings. **d** Lasing spectrum for a network without (top) and with (bottom) a refractive index shift of the red link, as schematised in the inset

simulations (see Supplementary Fig. 1). Such losses would be largely reduced if the network was composed of a higher refractive index material, such as GaAs or InP, nanostructured by lithography. While the presence of homogenous loss can change the gain available, its main effect is to increase the lasing threshold. Instead, a non-homogeneous loss can potentially suppress or enhance a few specific lasing modes, as shown in Fig. 5. The theoretical treatment of light scattering at the node could be further improved by including a scattering matrix, as shown in ref. [19], but at the price of longer computational times.

In conclusion, we demonstrated random lasing from a polymer nanophotonic network and proposed a graph approach to model its photonic properties. The network architecture promotes lasing of the most confined optical modes, which are shown to be spatially localised. The network is planar and therefore compatible with conventional semiconductor laser production. Nanophotonic networks made of inorganic semiconductor gain materials could be electrically pumped, upon placing electrical contacts at the nodes. Network random lasers can be easily extended to networks connected in three dimensions or planar networks with designed topology, and the light localisation length and threshold can be tuned with the network topology to further control the lasing action. The network lasers we have proposed are promising for sensitive sensors due to the high surface area, narrow linewidth and mode localisation, as well as for on-chip laser sources when outcoupled to external waveguides integrated on chip. Besides lasers, we envisage subwavelength photonic networks will bring a powerful way to control light flow and localisation for future classical and quantum technologies.

## Methods

**Fabrication of polymer fibre networks**. Polymethyl methacrylate (PMMA) was dissolved in a mixture of chloroform and dimethyl sulfoxide (DMSO) (volume ratio 9:2)[48,49]. Rhodamine-6G was then added to the solution with a concentration of 1% wt:wt relative to the polymer matrix. The solution was mixed by mechanical stirring and loaded in a 1 mL syringe tipped with a stainless steel needle. Electrospinning was performed by applying a bias (10–15 kV, EL60R0.6-22, Glassman High Voltage) between the needle and a $10 \times 10 \, cm^2$ Cu plate positioned 10 cm away, while injecting the solution at a constant flow rate (0.5–1 mL h$^{-1}$) using a syringe pump (Harvard Apparatus). Free-standing fibre networks were deposited on TEM grids with $425 \times 425 \, \mu m$ opening (TAAB Laboratories Equipment Ltd.). After deposition, samples were stored in a glovebox (Jacomex, GP[Concept]) and annealed at 80 °C for 5 min in nitrogen atmosphere to favour the formation of fibre joints at the nodes of the network, without degrading the emission properties of the embedded chromophores.

**Optical measurements**. Samples were optically pumped at room-temperature with a $\lambda = 532 \, nm$ pulsed laser (TEEM Microchip, pulse width 500 ps, spot diameter ~160 μm). The emission was spectrally analysed using a grating spectrometer (Princeton Instruments Isoplane-320) equipped with a 1800 gr mm$^{-1}$ holographic grating (0.05 nm resolution) and CCD camera (Princeton Instruments Pixis 400). For hyperspectral imaging, a lens in front of the spectrometer was mounted on a translating stage attached to a step motor. The lens was scanned in 25 μm steps and spectral data was recorded for each lens position, resulting in a 3D data set (one spatial dimension × wavelength × lens position). Hyperspectral images were reconstructed from the data set by selecting a particular wavelength of interest and spectra from the entire scanning area was extracted by summing the 3D data set across the two spatial dimensions. Mode control experiments by patterned pumping were carried out using a digital micromirror device (DMD, Ajile AJD-4500), which was inserted in the incoming beam pathway. The illumination pattern was then specified by the pattern projected on the DMD.

**Modelling of network modes**. Modes in the photonic network were modelled by solving Maxwell's equations on a graph. Light propagation along the (single-mode) link from node $i$ to node $j$ of length $L_{ij}$ is described by phase acquired by the electric field $E(x_j) = E(x_i)e^{ikL_{ij}}$, and multiple scattering process is described as a boundary-condition problem for the waves meeting at the nodes. Solutions are found by enforcing Neumann boundary conditions[35] $E_i = E_j$, $\sum dE_i/dx = 0$, which ensures that energy is conserved at the scattering node. Losses are only considered at the network boundary and out-of-plane scattering losses at nodes are ignored. Further details on numerical implementation of the model is provided in Supplementary Note 2. The wavelength shift due to refractive index perturbation was calculated by modifying the link lengths of the graph. For global change in index, the length $L_{ij}$ of

all links was increased by a factor $\Delta n$ (relative index change), whereas for local change, only the length of one particular link was increased. The model was then used to calculate the modes of the modified graph. The spectrum was obtained by defining a Lorentzian function for each mode, centred at Re($k$) and linewidth given by |Im($k$)|.

## Data availability

Data is publicly available in Figshare[50]. The code is available on request.

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

## Acknowledgements

We would like to thank Maria Moffa for sample preparation, Yohann Machu for modelling Buffons graph and Juan José Saenz for fruitful discussions. The research leading to these results has received funding from the Engineering and Physical Sciences Research Council (EPSRC), the Leverhulme Trust, the Royal Society, the European Research Council under the European Union's Seventh Framework Programme (FP/2007-2013)/ERC Grant Agreement no. 306357 (ERC Starting Grant "NANO-JETS").

## Author contributions

R.S. conceived the experiments, M.G., J.B. and R.S. developed the model, A.C. and D.P. provided the samples, M.G. and D.S. conducted the experiments and analysed the results. All authors reviewed the manuscript, discussed results and contributed to the manuscript preparation.

## Additional information

**Competing interests:** The authors declare no competing interests.

