## [Peer Review File · Nature Communications]

Reviewers' comments:

Reviewer #1 (Remarks to the Author):

I have carefully read the manuscript NCOMMS-18-22455.

The subject of the paper is a report on measurements and calculations on random lasing effects in a 10^{-2} mm² size sample of a planar network of polymeric nanofibres (whose diameter is smaller than the wavelength of the light with which they are excited and are, thus, in the single mode transport regime).

The strength of this work lies on the fact that although similar lasing effects in random systems have been reported before in networks (and in some cases discussed in terms of graphs) the authors provide clear and convincing evidence that the effects here go well beyond the single scattering regime, and show collective effects stemming from localized modes in small regions of the sample, in a sample that that can be obtained with very easy fabrication techniques.

In particular the clear experimental signature (see figure 2 f) of the distribution of mode spacing obtained through the hyperspectral imaging, well in agreement with theoretical calculations is one of the strong points of this work, together with the images (figures 2 a b c) of the lasing activity in different lasing modes.

The connection established numerically between the lasing threshold of the different modes and their localization in the interior or close to the periphery of the sample is a very important point of this paper (see figure 3b).

A few observations that the authors might consider, that will improve the usefulness of this article

Not many in the readership of this article and particularly in the community of nano-optics to which it will be of great interest are fully conversant on the literature or sometimes even the terminology of networks, so it would be of great use to add to the paper some basic bibliographical references beyond the more technical ones already provided, which are more illustrations of specific characteristics of networks.

Since the statistical analysis of the mode spacing distribution is an important part of this work, and the finite size effects of a small sample are clearly an issue, as the discussion about the localized modes on the edge indicates (see fig 3 and the corresponding discussion) it might be useful if the issue of mode spacing in small samples can be clarified further, and perhaps moving to the main body of the paper the useful information and figure in section D of the supplemental material.

In the first paragraph of the results a number is estimated for the gain length (10 microns) and it would be useful to expand on how this estimate is done, particularly since it plays an important role in the section of Discussion in page 7.

The authors make the very interesting point that in a nanophotonic network the process of light scattering can be controlled via the connectivity of the network, while the transport process depends on link size, opening the possibility, in principle to control these two characteristics in a decoupled way. It would be interesting to provide some information about how can this be achieved in the fabrication process (as opposed to the analytical calculations in figure 4 where this is illustrated).

The authors provide evidence of the magnitude of losses to out of plane scattering (through FDTD analysis) in an X-crossing, and presumably this could be even more important in higher

connectivity crossings, as some of the ones in the experimental samples. The authors correctly point out that higher index materials could make these losses smaller, but it would be useful to expand a bit on the discussion about the specific ways in which these losses could affect the experimental results in figure 2, since they are ignored in the theoretical/numerical calculations in figures 3 and 4.

When the authors describe the structure of the networks considered, they should make the notation a bit clearer. See for example the caption of figure 3 which says 143 nodes, 293 links, degree 6). I assume that here the word degree means the degrees of the nodes which are not close to the periphery of the sample; those nodes clearly have a degree smaller than six. If by degree one means the average degree in the sample, then it should be close to 4.1, twice the number of links divided the number of nodes, as in elementary network theory, and it is six only in this restricted sense. Small point but it should be clarified.

The point at the end of page 6 that just the increase on total length associated with the increase in the degree is not enough to explain the threshold degree, and a stronger onset of collective effects leading to localization has to be operative is a very important observation, and perhaps it could be expanded a little.

In summary, in spite of some of the midl criticisms that I mention above, I think this is an extremely useful paper which certainly deserves publication, and will be of great interest both to the communities of localization of light, nano photonics, and networks.

I think that after the authors had a chance to address some of the issues mentioned above this paper should be published.

Reviewer #2 (Remarks to the Author):

The paper submitted to Nature Communications by Michele Gaio et al. contains a theoretical and experimental study of random lasing on a graph network. Multi-mode lasing is demonstrated on this novel system and compared to theoretical models.

As a referee my opinion is somewhat divided as the manuscript contains both very appealing and some weak points. I will list them in the following for the further consideration of the editor and of the authors:

On the positive side, I first want to stress that the graph-based nano-photopic laser introduced here seems to be a novel and interesting platform and the authors certainly invested considerable work into its experimental fabrication. Also the fact that the laser modes can be nicely visualized on the graph is a very appealing aspect, in particular as it allows one to get direct access to the underlying lasing operation. This is quite an unusual aspect, especially for a random laser, where the internal structure of modes is typically out of reach experimentally. Also the direct observation of the change in lasing threshold with the topology of the underlying graph is a very nice aspect.

On the downside, it also needs to be said here that there are already quite a few realisations of random lasers around, in particular also such based on networks (see refs. 36-40 in the manuscript). I would thus have appreciated to see how this new realisation solves one or several of the problems that random lasers typically have (as, e.g., their limited controllability) or an intriguing fundamental aspect related to the topological features of the laser (going beyond the change in threshold alone). Alternatively, it would have been wonderful to see already a proof-of-principle demonstration of how this laser can, indeed, be used for those applications that the

authors mention in their conclusion (sensing, information processing etc.).

In the following, I also list a few technical points that should be clarified:

+ The definition of a mesh topology and of its degree should be properly introduced.

+ If I understood the manuscript correctly, the authors compare the statistics of the experimentally determined lasing frequencies with the level statistics of a random network model. I don't think these two quantities can be directly compared as the lasing modes above threshold also interact with each other in a non-linear manner (see also the comment in the summary of the paper on "strong mode competition"). Since this non-linear competition is not present in the random network model (correct?), the statistics will also be different for these two cases (especially for modes that are spectrally very close).

+ My impression is that the level statistics involved in this network (in particular for the lasing thresholds and for the inverse participation ratios) should be described by the Euclidian Random Matrix Theory or a suitably adjusted variant thereof - see, e.g., the following paper and references therein: A. Goetschy and S. E. Skipetrov, Phys. Rev. E 84, 011150 (2011).

+ The authors mention in the conclusion that their laser can also be electrically pumped. Can the authors please explain this in more detail?

In summary, I think that this manuscript contains several nice features, but also a number of weaknesses that prevent me from expressing my unequivocal support for publication in Nature Communications.

Reply to Reviewer #1

We thank the Reviewer for the constructive comments summarized by the judgement that this “*is an extremely useful paper which certainly deserves publication*”. We provide a point-to-point reply here below.

I have carefully read the manuscript NCOMMS-18-22455.

The subject of the paper is a report on measurements and calculations on random lasing effects in a 10^{-2}mm^2 size sample of a planar network of polymeric nanofibres (whose diameter is smaller than the wavelength of the light with which they are excited and are, thus, in the single mode transport regime).

The strength of this work lies on the fact that although similar lasing effects in random systems have been reported before in networks (and in some cases discussed in terms of graphs) the authors provide clear and convincing evidence that the effects here go well beyond the single scattering regime, and show collective effects stemming from localized modes in small regions of the sample, in a sample that that can be obtained with very easy fabrication techniques.

In particular the clear experimental signature (see figure 2 f) of the distribution of mode spacing obtained through the hyperspectral imaging, well in agreement with theoretical calculations is one of the strong points of this work, together with the images (figures 2 a b c) of the lasing activity in different lasing modes.

The connection established numerically between the lasing threshold of the different modes and their localization in the interior or close to the periphery of the sample is a very important point of this paper (see figure 3b).

A few observations that the authors might consider, that will improve the usefulness of this article

1. *Not many in the readership of this article and particularly in the community of nano-optics to which it will be of great interest are fully conversant on the literature or sometimes even the terminology of networks, so it would be of great use to add to the paper some basic bibliographical references beyond the more technical ones already provided, which are more illustrations of specific characteristics of networks.*

We have added a reference to the book by Estrada and Knight “A First Course in Network Theory” Oxford University Press 2015, to complement Reference 21 which are basic texts and illustrate the specific characteristics of networks.

2. *Since the statistical analysis of the mode spacing distribution is an important part of this work, and the finite size effects of a small sample are clearly an issue, as the discussion about the localized modes on the edge indicates (see fig 3 and the corresponding discussion) it might be useful if the issue of mode spacing in small samples can be clarified further, and perhaps moving to the main body of the paper the useful information and figure in section D of the supplemental material.*

We have added the figure in section D as an inset in Fig. 3f of the main text and have included a more detailed description:

“Upon simulating larger networks, the level spacing distribution clearly converges towards an exponential behavior, as shown in the inset of Fig. 3f, when the network is larger than the mode localization length.”

3. *In the first paragraph of the results a number is estimated for the gain length (10 microns) and it would be useful to expand on how this estimate is done, particularly since it plays an important role in the section of Discussion in page 7.*

We have added a brief explanation in the main text:

“The gain length is estimated to be 10 μm , which is calculated from the stimulated emission cross-section of the dye by $L_g = (\Pi \sigma)^{-1}$ (see Supplementary Note 7).”

And have added details of the calculation in Supplementary Information:

“The gain length of Rh6G (L_g) is estimated from its density in polymer (1% by weight) and its stimulated emission cross-section ($\sigma = 3 \times 10^{-20} \text{ m}^2$). The density of the polymer is $\sim 1 \text{ g cm}^{-3}$ and of Rh6G is 479 g mol^{-1} , so the density of Rh6G in polymer (Π) is:

$$\begin{aligned}\Pi &= (0.01 \times 1 \text{ g cm}^{-3} / 479 \text{ g mol}^{-1}) \times 6 \times 10^{23} \text{ molecules mol}^{-1} \\ &= 1.25 \times 10^{25} \text{ molecules m}^{-3}\end{aligned}$$

Gain length is thus:

$$\begin{aligned}L_g &= (\Pi \sigma)^{-1} \\ &= 1 / (1.25 \times 10^{25} \text{ m}^{-3} \times 3 \times 10^{-20} \text{ m}^2) \\ &= 4 \mu\text{m}\end{aligned}$$

However, this estimate is a best case scenario as it neglects losses and assumes perfect confinement of the mode inside the fibre. By taking into account the overlap of the fundamental mode with the fibre, which varies from 0.4 - 0.8 for fibre diameters between 200 - 500 nm, the gain length is estimated to be 5-10 μm .”

4. *The authors make the very interesting point that in a nanophotonic network the process of light scattering can be controlled via the connectivity of the network, while the transport process depends on link size, opening the possibility, in principle to control these two characteristics in a decoupled way. It would be interesting to provide some information about how can this be achieved in the fabrication process (as opposed to the analytical calculations in figure 4 where this is illustrated).*

Following the Referee’s comment, in the revised manuscript we added additional information about nanofabrication methods that would allow for controlling the number of nodes and the nanofibre length independently, thus enhancing the generation of nanophotonic networks with controlled topology. Such techniques include near-field electrospinning (Ref. Sun2006), which, in contrast with conventional electrospinning methods, exploits the stable region of the electrospun jet close to the metallic needle (typical extension of the stable region is of the order of 1 mm) for the deposition of individual extruded filaments, with diameters down to 100 nm, in a controlled way and following pre-designed elementary architectures. Nanophotonic networks can be also fabricated by printing technologies, among them the direct ink writing (Ref. Lewis2006) is the most promising for printing filaments with size down to 600 nm. A different approach can be also pursued by using soft nanolithographies (Ref. Xia1998). The following sentence and the new References Sun2006, Lewis2006, Xia1998 have been added to the revised manuscript:

“Nanofabrication methods such as near-field electrospinning^{Sun2006}, direct ink writing^{Lewis2006} and soft lithographies^{Xia1998} would in principle allow for highly controlled deposition of the individual waveguides, with pre-defined and independently controlled connectivity and size.”

5. *The authors provide evidence of the magnitude of losses to out of plane scattering (through FDTD analysis) in an X-crossing, and presumably this could be even more important in higher connectivity crossings, as some of the ones in the experimental samples. The authors correctly point out that higher index materials could make these losses smaller, but it would be useful to expand a bit on the discussion about the specific ways in which these losses could affect the experimental results in figure 2, since they are ignored in the theoretical/numerical calculations in figures 3 and 4.*

We have clarified how losses affect our results in the discussion and have added the following sentences:

“While the presence of homogenous loss can change the gain available, its main effect is to increase the lasing threshold. Instead, a non-homogenous loss can potentially suppress or enhance a few specific lasing modes, as shown in Fig. 5.”

6. *When the authors describe the structure of the networks considered, they should make the notation a bit clearer. See for example the caption of figure 3 which says 143 nodes, 293 links, degree 6). I assume that here the word degree means the degrees of the nodes which are not close to the periphery of the sample; those nodes clearly have a degree smaller than six. If by degree one means the average degree in the sample, then it should be close to 4.1, twice the number of links divided the number of nodes, as in elementary network theory, and it is six only in this restricted sense. Small point but it should be clarified.*

We have clarified this in the main text, that the degree is calculated for “*the nodes which are not close to the periphery of the sample*” both in the caption of Fig. 3 and in the introduction:

“Each structure is a planar disordered network with a partial mesh topology, where all nodes connect directly and non-hierarchically to their nearest neighbour. The network has degree D , defined as the number of connections of each node but excluding the ones on the periphery, in the range 4 to 6 (average degree is 4.4) ...”

Fig. 3 Caption:

“For a given network topology (143 nodes, 293 links, average degree $D=6$ (excluding nodes at the periphery) ...”

7. *The point at the end of page 6 that just the increase on total length associated with the increase in the degree is not enough to explain the threshold degree, and a stronger onset of collective effects leading to localization has to be operative is a very important observation, and perhaps it could be expanded a little.*

An increase of the degree means that the network has more links, and therefore more total gain available, which is proportional to the total number of molecules, i.e. the network total length. In our case the network length doubles when going from $D=3$ to $D=6$, but the threshold diminishes by much more, by a factor of 10.

We have clarified the sentence in the text:

“We point out that the increase in network length with increasing degree, which is an increase in the total gain available, cannot explain the threshold decrease alone. The total gain increases by a factor of ~ 2 (when comparing $D = 3$ and $D = 6$) while the threshold reduces by a factor of ~ 10 . ”

In summary, in spite of some of the mild criticisms that I mention above, I think this is an extremely useful paper which certainly deserves publication, and will be of great interest both to the communities of localization of light, nano photonics, and networks.

I think that after the authors had a chance to address some of the issues mentioned above this paper should be published.

Reply to Reviewer #2

We thank the Reviewer for the constructive comments.
We provide a point-to-point reply here below.

The paper submitted to Nature Communications by Michele Gaio et al. contains a theoretical and experimental study of random lasing on a graph network. Multi-mode lasing is demonstrated on this novel system and compared to theoretical models.

As a referee my opinion is somewhat divided as the manuscript contains both very appealing and some weak points. I will list them in the following for the further consideration of the editor and of the authors:

On the positive side, I first want to stress that the graph-based nano-photonic laser introduced here seems to be a novel and interesting platform and the authors certainly invested considerable work into its experimental fabrication. Also the fact that the laser modes can be nicely visualized on the graph is a very appealing aspect, in particular as it allows one to get direct access to the underlying lasing operation. This is quite an unusual aspect, especially for a random laser, where the internal structure of modes is typically out of reach experimentally. Also the direct observation of the change in lasing threshold with the topology of the underlying graph is a very nice aspect.

1. *On the downside, it also needs to be said here that there are already quite a few realisations of random lasers around, in particular also such based on networks (see refs. 36-40 in the manuscript). I would thus have appreciated to see how this new realisation solves one or several of the problems that random lasers typically have (as, e.g., their limited controllability) or an intriguing fundamental aspect related to the topological features of the laser (going beyond the change in threshold alone). Alternatively, it would have been wonderful to see already a proof-of-principle demonstration of how this laser can, indeed, be used for those applications that the authors mention in their conclusion (sensing, information processing etc.).*

We thank the Referee for her/his comment that gives us the opportunity to better clarify a few important aspects about the novelty of our work. Differently from our work, devices shown in Refs. 36-40 are not random lasers that can be parameterized on a graph in the sense of this work, as in those samples waveguiding along the active bodies, and three-dimensional scattering effects cannot be decoupled and separately controlled, which is the most important aspect of our networks and graph models. This is, to our knowledge, the first reported laser where light is confined in the complex graph network, and therefore where the optical modes are unequivocally designed by the network topology.

As suggested by the Reviewer, we have added a figure to the manuscript, to highlight the sensitivity of the network localised lasing modes, which can be used for spectral shaping and sensing. In the new Fig. 5, the pump illumination profile is varied to a checkerboard, and the lasing output is drastically changed. We explain that this experimental result is due to spatially-dependent modal gain, which favours a mode against another. In terms of the network topology, the use of a checkerboard pumping beam decreases the average length of the active links (i.e. those optically pumped which can sustain stimulated emission), which also indicates the possibility to vary the spectral emission via the network parameters.

Furthermore, the network is quite sensitive to local changes of the refractive index. In Fig. 5c we simulate how the modes of the network shift when the refractive index of a single link is changed. For a change as small as 10^{-2} , we can see a clear shift in some of the lasing modes, in particular for those that are localised around that specific link. Again, this high sensitivity is due to the localised nature of the lasing modes and highlights the potential use of network lasers for sensing, thus meeting the proof-of-principle demonstration suggested by the Referee.

The following section and figure have been added in the main text:

“The localised modes of the networks are very sensitive to *local* perturbations, either in the form of non-uniform pumping or as induced by a local change in refractive index, mimicking the binding of a target analyte. Fig. 5 describes these two effects. In Fig. 5a-b, a non-uniform pump profile, with the shape of a checkerboard, modifies the lasing profile by supressing or enhancing a few lasing modes. This experimental result is due to spatially-dependent modal gain, which favours a mode against another and which is very efficient because of mode localisation^{Bachelard (2014) and S.F. Liew (2014)}.

Moreover, the network is very sensitive to a local change of the refractive index. In Fig. 5c-d we plot how the modes of the network shift when the refractive index of a single link is changed. For a change as small as $\Delta n = 10^{-2}$, we can see a clear shift in some of the lasing modes, those that are localised around that specific link. This gives us a refractive index sensitivity of 20 nm per RIU (refractive index units) or 1 nm per RIU per micrometre (link length 20 μm). In comparison, a global change in the refractive index across all links of the network (total length 4.834 mm) has a sensitivity of 570 nm per RIU (see Supplementary Figure 5) or 0.12 nm per RIU per micrometre. The network is thus 10 times more sensitive to local perturbations than global changes in refractive index.”

Figure 5. Spectral shaping and optical sensing. (a) Far-field image showing fluorescence from a network that is pumped with checkerboard pattern, with patches of size of $52 \mu\text{m} \times 60 \mu\text{m}$. (b) Measured lasing spectrum obtained when pumping the network with the checkerboard pattern (top), as well pumping it uniformly (bottom). (c) Calculated map of the lasing mode shift versus the refractive index shift of a selected link. A few modes shift, and mode coupling is evident in the avoided crossings. (d) Lasing spectrum for a network without (top) and with (bottom) a refractive index shift of the red link, as schematised in the inset.

In the following, I also list a few technical points that should be clarified:

2. *The definition of a mesh topology and of its degree should be properly introduced.*

We have clarified the definition of degree in the main text and added a sentence:

“Each structure is a planar disordered network with a partial mesh topology, where all nodes connect directly and non-hierarchically to their nearest neighbour. The network has *degree D*, defined as the number of connections of each node but excluding the ones on the periphery, in the range 4 to 6 (average degree is 4.4) ...”

3. *If I understood the manuscript correctly, the authors compare the statistics of the experimentally determined lasing frequencies with the level statistics of a random network model. I don't think these two quantities can be directly compared as the lasing modes above threshold also interact with each other in a non-linear manner (see also the comment in the summary of the paper on “strong mode competition”). Since this non-linear competition is not present in the random network model (correct?), the statistics will also be different for these two cases (especially for modes that are spectrally very close).*

My impression is that the level statistics involved in this network (in particular for the lasing thresholds and for the inverse participation ratios) should be described by the Euclidian Random Matrix Theory or a suitably adjusted variant thereof - see, e.g., the following paper and references therein: A. Goetschy and S. E. Skipetrov, Phys. Rev. E 84, 011150 (2011).

We agree with the Reviewer, the mode statistics comes from a random matrix theory for the passive system, which excludes mode competition. Our model does not include nonlinear mode competition, and it is valid only under the assumption of linear and undepleted gain as we mention on page 5. We have included the suggested reference (see below).

As the Reviewer points out, the mode interaction could lead to a change of the lasing spectrum, and therefore of the mode spacing statistics. We believe that the nonlinear spectral interaction is very weak in our experiments, as we are near the lasing threshold. This is proved by figure S3 where we report the spectral evolution as a function of pump power. The lasing modes evolve as vertical lines, indicating that their spectral position is unchanged for increasing excitation power. We have added the following paragraph in the manuscript to explain this in detail.

“We note that random matrix theory description is valid for a passive system^{Goetschy2011} and so the mode interaction above threshold could lead to a change in the lasing spectrum, and therefore of the mode spacing statistics. However, we operate close to the lasing threshold, and as shown in Supplementary Figure 2, the lasing peaks do not shift in frequency with pump power.”

4. *The authors mention in the conclusion that their laser can also be electrically pumped. Can the authors please explain this in more detail?*

Laser emission upon electrical pumping could be achieved in networks made with inorganic semiconductor gain material. For instance, networks could be fabricated on typical semiconductor epitaxial wafers that are used for ridge-waveguide laser diodes, and electrical injection could be achieved by placing electrical contacts at the nodes of the network.

To explain how electrical pumping could be achieved, we have added the following sentence to the main text:

“The network is planar and therefore compatible with conventional semiconductor laser production. Nanophotonic networks made of inorganic semiconductor gain materials could be electrically pumped, upon placing electrical contacts at the nodes.”

Reviewer #1 (Remarks to the Author):

I have carefully considered the rebuttal letter of the authors, in which they answered, clarified (and in a couple of cases corrected) the points I raised in my previous review, and also their comments and corrections in reply to a few very valid points raised by the other reviewer.

I think that in balance they have addressed correctly both of our concerns, and, thus, I support the publication of the new updated form of this manuscript, in the way it is now.

Reviewer #2 (Remarks to the Author):

The authors have resubmitted an improved manuscript in which an attempt was made to incorporate the suggestions of both reviewers.

While I do see that there is still room for developing these nano-photonics graph-lasers into more controllable light source, I am willing to acknowledge that the contribution made in this paper makes a first important step into this direction and recommend it for publication in Nature Communications.

My only remaining concern is the following: the authors emphasize in the manuscript text that their modes follow the statistics of RMT below threshold, which suggests that non-linear mode-competitions are negligible. At the same time the following statement is made in the conclusions, which seems to contradict this analysis: "The network lasers we have proposed are promising for [...] strong mode competition..." This point should be clarified before publication.

REPLY TO REVIEWERS' COMMENTS:

We thank the Reviewers for their time and constructive comments. Below is a point-to-point reply addressing the comments.

Reviewer #1 (Remarks to the Author):

I have carefully considered the rebuttal letter of the authors, in which they answered, clarified (and in a couple of cases corrected) the points I raised in my previous review, and also their comments and corrections in reply to a few very valid points raised by the other reviewer.

I think that in balance they have addressed correctly both of our concerns, and, thus, I support the publication of the new updated form of this manuscript, in the way it is now.

We are thank Reviewer 1 for carefully considering our revised manuscript.

Reviewer #2 (Remarks to the Author):

The authors have resubmitted an improved manuscript in which an attempt was made to incorporate the suggestions of both reviewers.

While I do see that there is still room for developing these nano-photonics graph-lasers into more controllable light source, I am willing acknowledge that the contribution made in this paper makes a first important step into this direction and recommend it for publication in Nature Communications.

My only remaining concern is the following: the authors emphasize in the manuscript text that their modes follow the statistics of RMT below threshold, which suggests that non-linear mode-competitions are negligible. At the same the following statement is made in the conclusions, which seems to contradict this analysis: "The network lasers we have proposed are promising for [...] strong mode competition..." This point should be clarified before publication.

We agree with the reviewer that we can describe our system with below-threshold theory, excluding nonlinear interactions, and therefore we have modified the sentence in the concluding paragraph:

*"The network lasers we have proposed are promising for sensitive sensors due to the high surface area, narrow linewidth and **mode localisation**..."*